# ROBUST MODEL-BASED OPTIMIZATION FOR CHALLENGING FITNESS LANDSCAPES

**Saba Ghaffari**[1*]    **Ehsan Saleh**[1*]    **Alexander G. Schwing**[1]    **Yu-Xiong Wang**[1]
**Martin D. Burke**[1]    **Saurabh Sinha**[2]
[1]University of Illinois Urbana-Champaign, [2]Georgia Institute of Technology
`{sabag2, ehsans2, aschwing, yxw, mdburke}@illinois.edu`,
`saurabh.sinha@bme.gatech.edu`

## ABSTRACT

Protein design, a grand challenge of the day, involves optimization on a fitness landscape, and leading methods adopt a model-based approach where a model is trained on a training set (protein sequences and fitness) and proposes candidates to explore next. These methods are challenged by sparsity of high-fitness samples in the training set, a problem that has been in the literature. A less recognized but equally important problem stems from the distribution of training samples in the design space: leading methods are not designed for scenarios where the desired optimum is in a region that is not only poorly represented in training data, but also relatively far from the highly represented low-fitness regions. We show that this problem of "separation" in the design space is a significant bottleneck in existing model-based optimization tools and propose a new approach that uses a novel VAE as its search model to overcome the problem. We demonstrate its advantage over prior methods in robustly finding improved samples, regardless of the imbalance and separation between low- and high-fitness training samples. Our comprehensive benchmark on real and semi-synthetic protein datasets as well as solution design for physics-informed neural networks, showcases the generality of our approach in discrete and continuous design spaces. Our implementation is available at https://github.com/sabagh1994/PGVAE.

## 1 INTRODUCTION

Protein engineering is the problem of designing novel protein sequences with desired quantifiable properties, e.g., enzymatic activity, fluorescence intensity, for a variety of applications in chemistry and bioengineering (Fox et al., 2007; Lagassé et al., 2017; Biswas et al., 2021). Protein engineering is approached by optimization over the protein fitness landscape which specifies the mapping between protein sequences and their measurable property, i.e., fitness. It is believed that the protein fitness landscape is extremely sparse, i.e., only a minuscule fraction of sequences have non-zero fitness, and rugged, i.e., peaks of "fit" sequences are narrow and separated from each other by deep valleys (Romero & Arnold, 2009), which greatly complicates the problem of protein design. Directed evolution is the most widely adopted technique for sequence design in laboratory environment (Arnold, 1998). In this greedy *local* search approach, first a set of variants of a naturally occurring ("wild type") sequence are tested for the desired property, then the variants with improved property form the starting points of the next round of mutations (selected uniformly at random) and thus the next round of sequences to be tested. This process is repeated until an adequately high level of desired property is achieved. Despite advances, this strategy remains costly and laborious, prompting the development of model-guided searching schemes that support more efficient exploration of the sequence space (Biswas et al., 2018; Brookes & Listgarten, 2018; Gómez-Bombarelli et al., 2018; Brookes et al., 2019; Angermueller et al., 2019; Sinai et al., 2020; Ren et al., 2022). In particular, there is emerging agreement that optimization schemes that utilize ML models of the sequence-fitness relationship, learned from training sets that grow in size as the optimization progresses, can furnish better candidates for the next round of testing, and thus accelerate optimization, as compared to model-free approaches such as Bayesian optimization (Mockus, 2012; Sinai et al., 2020). Our work belongs to this genre of model-based optimization for sequence-function landscapes.

---

[*]Both authors contributed equally.

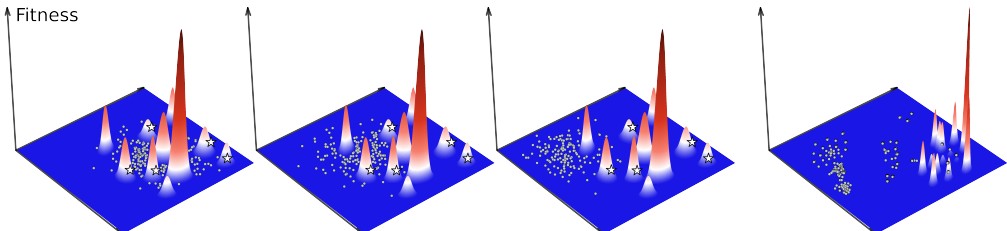

Figure 1: **Challenges of imbalance and separation in fitness landscape.** Each plot shows a sequence space (x-y plane) and fitness landscape (red-white-blue gradient), along with training data composition (white circles and stars). (A-C, left to right) In each of these hypothetical scenarios, sparsity of high-fitness training samples (white stars) relative to low-fitness samples (white circles), also called "imbalance" presents a challenge for MBO. Moreover, panel C shows a greater degree of separation between low- and high-fitness samples, compared to B and A, presenting significant additional challenge for MBO, above and beyond that due to imbalance. The rightmost panel is the schematic representation of real-world dataset of enzyme variants designed for an unnatural substrate (xyz) distinct from the substrate of the wild-type enzyme (xyz). The dataset comprises a few non-zero fitness variants (stars) that are far from the bulk of training samples, which have zero fitness (white circles). Hypothetical peaks have been drawn at the rare non-zero fitness variants, to illustrate that the fitness landscape presents the twin challenges of imbalance and separation, similar to that in panel C.

Intuitively, the success of fitness optimization depends on the extent to which functional proteins are represented in the experimentally derived data (training set) so that the characteristics of desired sequences can be inferred from them. Prior work has examined this challenge of "sparsity" in fitness landscapes, proposing methods that use a combination of "exploration" and "exploitation" to search in regions of the space less represented in training data (Romero et al., 2013; Gonzalez et al., 2015; Yang et al., 2019; Hie & Yang, 2022). Optimization success also depends on the distribution of training samples in the sequence space, in particular on whether the desired functional sequences are proximal to and easily reachable from the frequent but low-fitness training samples. This second challenge of "separation" (between the optima and training samples) in fitness landscape is relatively unexplored in the literature. In particular, it is not known how robust current search methods are when the optimum is located in a region that is poorly represented in the training set *and* is located relatively far (or separated due to rugged landscape) from the highly represented regions Figure 1. A real-world example of this is the problem of designing an enzyme for an unnatural target substrate, starting from the wild-type enzyme for a related natural substrate. Most variants of the wild type enzyme are not functional for the target substrate, thus the training set is sparse in sequences with non-zero fitness; furthermore, the rare variants that do have non-zero activity (fitness) for the target substrate are located relatively far from the wild-type and its immediate neighborhood that forms the bulk of the training set Figure 1 (rightmost panel).

We study the robustness of model-guided search schemes to the twin challenges of imbalance and separation in fitness landscape. We explore for the first time how search algorithms behave when training samples of high fitness are rare and separated from the more common, low-fitness training samples. (Here, separation is in the design or sequence space, not the fitness space.) Furthermore, given a fixed degree of separation, we investigate how the imbalance between the low- and high-fitness samples in the training set affect the performance of current methods. A robust algorithm should have consistent performance under varying separation and imbalance.

To this end, we propose a new model-based optimization (MBO) approach that uses VAE (Kingma & Welling, 2014) as its search model. The latent space of our VAE is explicitly structured by property (fitness) values of the samples (sequences) such that more desired samples are prioritized over the less desired ones and have *higher* probability of generation. This allows robust exploration of the regions containing more desired samples, regardless of the extent of their representation in the train set and of the extent of separation between low- and high-fitness samples in the train set. We refer to the proposed approach as a "*Property-Prioritized Generative Variational Auto-Encoder*" (PPGVAE).

Our approach is designed with the goal of obtaining improved samples in less number of MBO steps (less sampling budget), as desired in the sequence design problem. Methods that rely on systematic exploration techniques such as Gaussian processes (Gómez-Bombarelli et al., 2018) may not converge in small number of rounds (Srinivas et al., 2009); a problem that is exacerbated by higher dimensionality of the search space (Frazier, 2018; Djolonga et al., 2013). In general, optimization

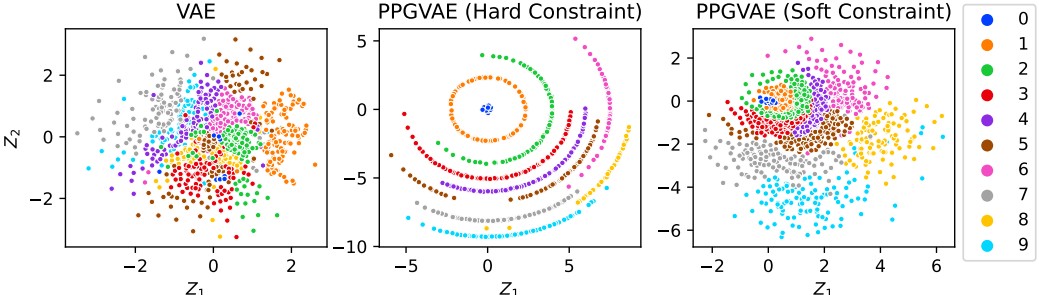

Figure 2: **Latent space of our PPGVAE vs Vanilla VAE.** PPGVAE and vanilla VAE were trained on a toy MNIST-derived dataset where property values decrease monotonically with digit value (zero has highest property value). Vanilla VAE (**Left**) scatters the rare samples of digit zero (blue) and samples of next-highest property value (digit one, orange) in the latent space, whereas PPGVAE (**Middle** and **Right**) maps digits with higher property values closer to the origin. This results in the classes with greatest property values having higher probability of generation. PPGVAE was run in two modes, where the relationship loss was enforced in a strong (**Middle**) or soft (**Right**) manner (see text).

with fewer MBO steps can be achieved by either 1) bringing more desired (higher fitness) samples closer together and prioritizing their exploration over the rest, as done in our approach, or 2) using higher weights for more desired samples in a weighted optimization setting (Brookes & Listgarten, 2018; Brookes et al., 2019; Gupta & Zou, 2019). Neither of these can be achieved by methods that condition the generation of samples on the the property values (Kang & Cho, 2018) or encode the properties as separate latent variables along with the samples (Guo et al., 2020; Chan et al., 2021). This is the key methodological gap in the state-of-the-art that is addressed by our new VAE technique for model-based optimization.

Through extensive benchmarking on real and semi-synthetic protein datasets we demonstrate that MBO with PPGVAE is superior to prior methods in robustly finding improved samples regardless of 1) the imbalance between low- and high-fitness training samples, and 2) the extent of their separation in the design space. Our approach is general and *not limited* to protein sequences, i.e., discrete design spaces. We further investigate MBO with PPGVAE on continuous designs spaces. In an application to physics-informed neural networks (PINN) (Raissi et al., 2019), we showcase that our method can consistently find improved high quality solutions, given PINN-derived solution sets overpopulated with low quality solutions separated from rare higher quality solutions. In section 2, MBO is reviewed. PPGVAE is explained in section 3 followed by experiments in section 4.

## 2  BACKGROUND

**Model Based Optimization.** Given $(x, y)$ pairs as the data points, e.g., protein sequence $x$ and its associated property $y$ (e.g., pKa value), the goal of MBO is to find $x \in \mathcal{X}$ that satisfy an objective $S$ regarding its property with high probability. This objective can be defined as maximizing the property value $y$, i.e., $S = \{y | y > y_m\}$ where $y_m$ is some threshold. Representing the search model with $p_\theta(x)$ (with parameters $\theta$), and the property oracle as $p_\beta(y|x)$ (with parameters $\beta$), MBO is commonly performed via an iterative process which consists of the following three steps at iteration $t$ (Fannjiang & Listgarten, 2020):

1. Taking $K$ samples from the search model, $\forall i \in \{1, ..., K\} \quad x_i^t \sim p_{\theta^t}(x)$;

2. Computing sample-specific weights using a monotonic function $f$ which is method-specific:
$$w_i := f(p_\beta(y_i^t \in S | x_i^t)); \qquad (1)$$

3. Updating the search model parameters via weighted maximum likelihood estimation (MLE):
$$\theta^{t+1} = \arg\max_\theta \sum_{i=1}^{K} w_i \log(p_\theta(x_i^t)). \qquad (2)$$

The last step optimizes for a search model that assigns higher probability to the data points satisfying the property objective $S$, i.e., where $p_\beta(y \in S | x)$ is high. Prior work by (Brookes & Listgarten,

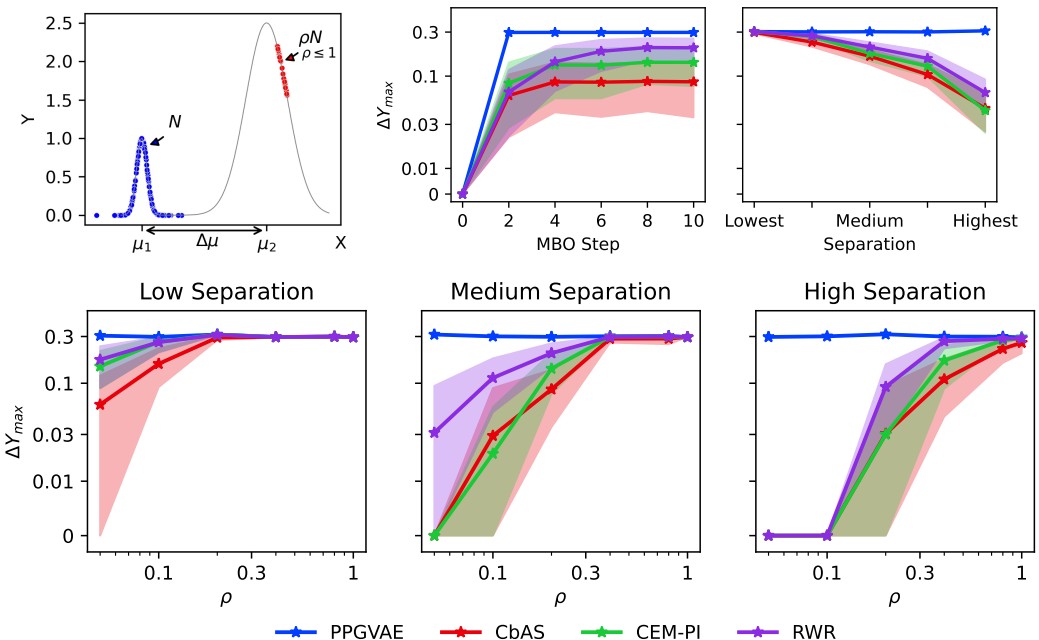

Figure 3: **Robustness to imbalance and separation in MBO for GMM.** A bimodal GMM is used as the property oracle (**top Left**), i.e., the fitness ($Y$) landscape on a one-dimensional sequence space ($X$). Separation is defined as the distance between the means of the two modes ($\Delta\mu$). Higher values of $\Delta\mu$ are associated with higher separation. Train sets were generated by taking $N$ samples from the less desired mode $\mu_1$ and $\rho N$ (imbalance ratio $\rho \leq 1$) samples from the more desired mode $\mu_2$. For a fixed separation, PPGVAE achieves robust relative improvement of the highest property sample generated ($\Delta Y_{max}$), regardless of the imbalance ratio (**Bottom panels**). Performance of PPGVAE, aggregated over all imbalance ratios, stays robust to increasing separation (**top Right**). PPGVAE converges in less number of MBO steps (**top Middle**).

2018) (DbAS) and (Brookes et al., 2019) (CbAS) have explored variants of weighting schemes for the optimization in the second step. Reward-weighted regression (RWR) (Peters & Schaal, 2007) and CEM-PI (Snoek et al., 2012) have additionally been benchmarked by CbAS, each providing a different weighting scheme. RWR has been used for policy learning in Reinforcement Learning (RL) and CEM-PI maximizes the probability of improvement over the best current value using the cross entropy method (Rubinstein, 1997; 1999).

Common to all these methods is that weighted MLE could suffer from reduced effective sample size. In contrast, our PPGVAE does not use weights. Instead, it assigns higher probability to the high fitness (more desired) data points by restructuring the latent space. Thus, allowing for the utilization of all samples in training the generative model (see Appendix A C).

**Exploration for Sequence Design.** In addition to weighting based generative methods, model-based RL (Angermueller et al., 2019) (Dyna PPO) and evolutionary greedy approaches (Sinai et al., 2020) (AdaLead) have been developed to perform search in the sequence space for improving fitness. More recently, (Ren et al., 2022) (PEX) proposed an evolutionary search that prioritizes variants with improved property which fall closer to the wild type sequence.

## 3    PROPERTY-PRIORITIZED GENERATIVE VARIATIONAL AUTO-ENCODER

To prioritize exploration and generation of rare, high-fitness samples, our PPGVAE uses property (fitness) values to restructure the latent space. The restructuring enforces samples with higher property to lie closer to the origin than the ones with lower property. As the samples with higher property lie closer to the origin, their probability of generation is higher under the VAE prior distribution $\mathcal{N}(0, I)$. Representing the encoder and its parameters with $Q$ and $\theta$, the structural constraint on $N$ samples is imposed by

$$\forall(\mu_\theta^i, \mu_\theta^j), \; i, j \in \{1, ..., N\} \quad \log(\Pr(\mu_\theta^i)) - \log(\Pr(\mu_\theta^j)) = \tau(y_i - y_j), \tag{3}$$

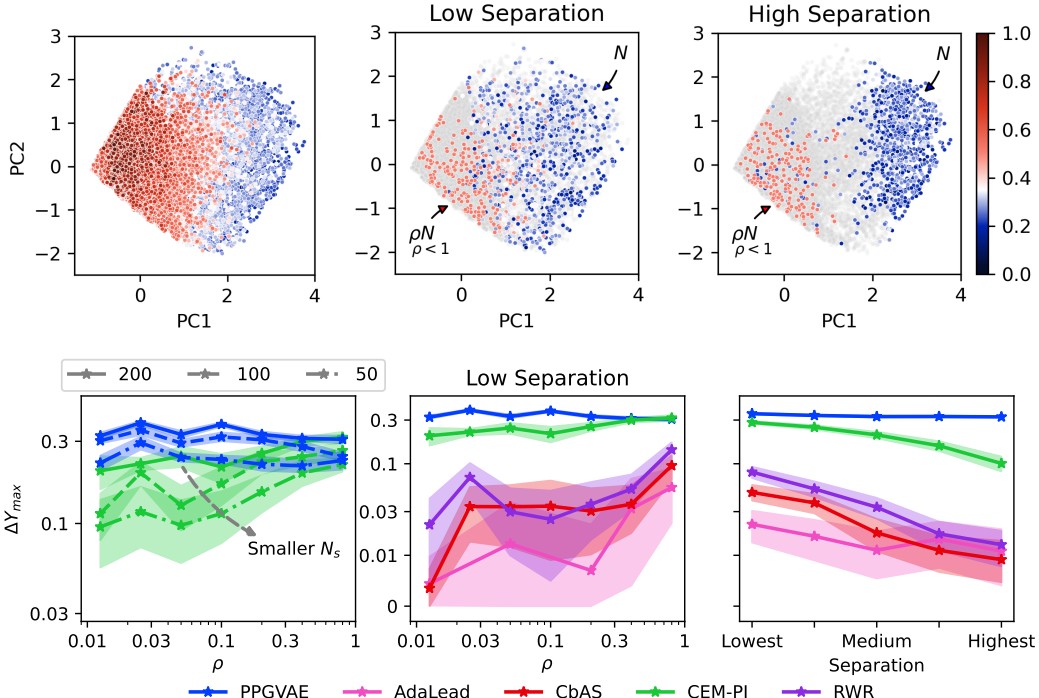

Figure 4: **Robustness to imbalance and separation in MBO for AAV dataset.** PCA plot for protein sequences in the dataset, colored with their property values (**top Left**). Blue and red color spectrum are used for less and more desired samples, respectively. Top middle and right panels show train sets with low and high separation, respectively, between the abundant less-desired and rare more-desired samples. PPGVAE achieves robust relative improvements (shown here for the low separation scenario), regardless of the imbalance ratio $\rho$ (**bottom Middle**). Its performance also stays robust to increasing separation (**bottom Right**). PPGVAE performance is only slightly affected by reducing its sampling budget per MBO step ($N_s$) (**bottom Left**).

where $\mu_\theta^i = Q_\theta(x_i)$ and $y_i$ are the latent space representation and property value of sample $x_i$, respectively. The probability of the encoded representation $\Pr(\mu_\theta^i)$ is computed w.r.t. the VAE prior distribution $\mathcal{N}(0, I)$ over the latent space, i.e., $\Pr(\mu_\theta^i) \propto \exp(\frac{-\mu_\theta^{i\,T}\mu_\theta^i}{2})$.

Intuitively, if higher values of property $y$ are desired, then $y_j \leq y_i$ results in sample $i$ getting mapped closer to the origin. This results in a higher probability of generating sample $i$ than sample $j$. The extent of prioritization between each pair of samples is controlled by the hyper-parameter $\tau$, often referred to as the temperature. The structural constraint is incorporated into the objective of a vanilla VAE as a relationship loss that should be minimized. This loss is defined as,

$$\mathcal{L}_r \propto \sum_{i,j} ||(\log(\Pr(\mu_\theta^i)) - \log(\Pr(\mu_\theta^j))) - \tau(y_i - y_j)||_2^2. \tag{4}$$

Combined with the vanilla VAE, the final objective of PPGVAE to be maximized is,

$$\mathbb{E}_{z \sim Q(.|x)}[\log(P(x|z)) - D_{\mathrm{KL}}(Q(z|x)\|P(z))] - \frac{\lambda_r}{\tau^2}\mathcal{L}_r, \tag{5}$$

where $\lambda_r$ is a hyper-parameter controlling the extent to which the relationship constraint is enforced. Here, we abused the notation and wrote $Q(z|x)$ for $\Pr(z|Q(x))$. To understand the impact of the structural constraint on the mapping of samples in the latent space, we define $q_i := \log(\Pr(\mu_\theta^i)) - \tau y_i$. Also, assume that the samples are independent and identically distributed (i.i.d.). Then minimizing the relationship loss can be rewritten as

$$\mathcal{L}_r \propto \mathbb{E}_{q_i, q_j}((q_i - q_j)^2) = \mathbb{E}_{q_i, q_j}(((q_i - \mathbb{E}(q_i)) - (q_j - \mathbb{E}(q_j)))^2). \tag{6}$$

Using the i.i.d. assumption this is simplified to,

$$\mathcal{L}_r \propto 2\mathrm{Var}(q_i) = 2\mathrm{Var}(\log(P(\mu_\theta^i)) - \tau y_i)) = 2\mathrm{Var}(-\frac{\mu_\theta^{i\,T}\mu_\theta^i}{2} - \tau y_i). \tag{7}$$

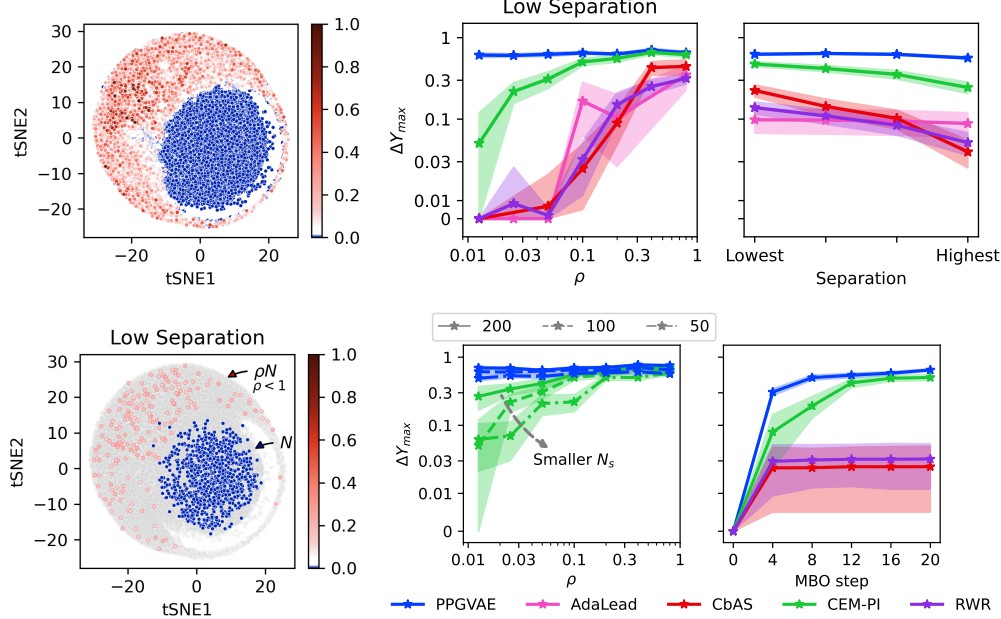

Figure 5: **Robustness to imbalance and separation in MBO for GB1 dataset.** The tSNE plot for the appended sequences of semi-synthetic GB1 dataset (**top Left**). Bottom left panel represents an example of train set for low separation between less and more desired samples, i.e., appended sequence of length three (see Figure A3 for an example of high separation). For a fixed separation level, PPGVAE provides robust improvements, regardless of the imbalance ratio (**top Middle**). It is also robust to the degree of separation, measured by aggregated performance over all imbalance ratios (**top Right**). PPGVAE has faster convergence (**bottom Right**) and achieves similar improvements with less sampling budget per MBO step ($N_s$) (**bottom Middle**).

Therefore, minimizing $\mathcal{L}_r$ is equivalent to minimizing the variance. This is equivalent to setting the random variable in RHS of Equation 7 to a constant value $C$; $\forall i: \quad \mu_\theta^{i\,T} \mu_\theta^i = 2(C - \tau y_i)$. This implies the distribution of samples with the same property value to be on the same sphere. The sphere lies closer to the origin for the samples with higher property values. This ensures that higher property samples have higher probability of generation under the VAE prior $N(0, I)$, while allowing for all samples to fully contribute to the optimization.

To demonstrate the impact of relationship loss on the latent space, PPGVAE, with a two-dimensional latent space, was trained on a toy MNIST dataset (Deng, 2012). The dataset contains synthetic property values that are monotonically decreasing with the digit class. Also, samples from the digit zero with the highest property are rarely represented (see Appendix D.3). Strong enforcement of the relationship loss, using a relatively large constant $\lambda_r$, aligns samples from each class on circles whose radius increases as the sample property decreases. Soft enforcement of the relationship loss, by gradually decreasing $\lambda_r$, makes the samples more dispersed, while roughly preserving their relative distance to the origin (Figure 2).

## 4 EXPERIMENTS

We will compare the performance of MBO with PPGVAE to the baseline algorithms that use generative models optimized with weighted MLE (CbAS, RWR, CEM-PI (Brookes et al., 2019)) for search in discrete (e.g., protein sequence) and continuous design spaces. For sequence design experiments, we additionally included AdaLead as a baseline.

In all optimization tasks, we 1) provide a definition of separation between less and more desired samples in the design space $\mathcal{X}$, 2) study the impact of varying imbalance between the representation of low- and high-fitness (property value) samples in the training set, given a fixed separation degree, and 3) study the impact of increasing separation. The ground truth property oracle was used in all experiments. The performance is measured by $\Delta Y_{\max}$ representing the relative improvement of the

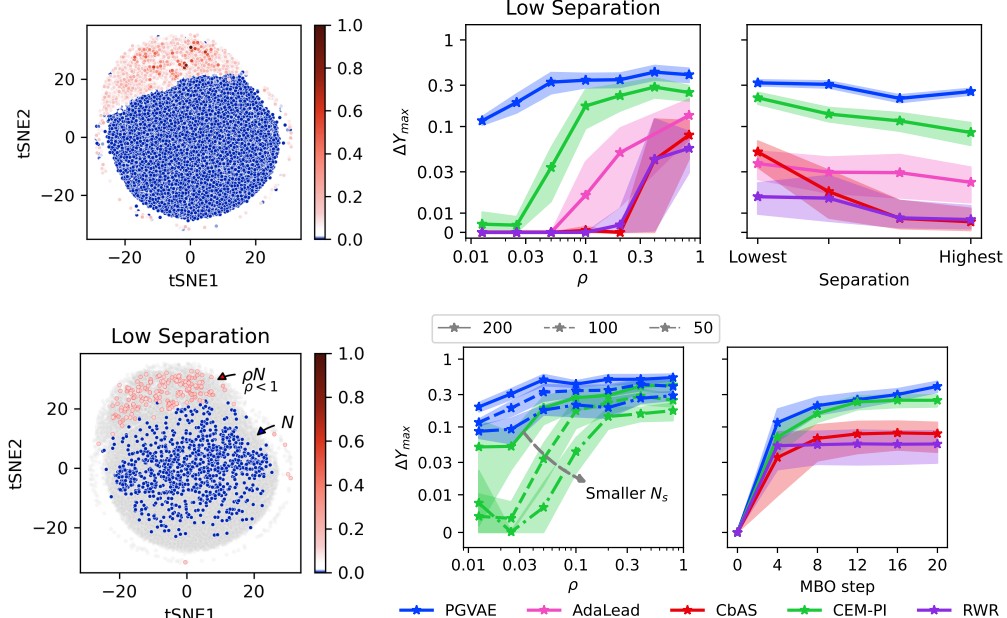

Figure 6: **Robustness to imbalance and separation in MBO for PhoQ dataset.** The panels and their corresponding observation have the same semantics as in Figure 5.

highest property found by the model to the highest property in the train set, i.e., initial set at the beginning of MBO (see Appendix D for further details).

## 4.1 GAUSSIAN MIXTURE MODEL

We use a bimodal Gaussian mixture model (GMM) as the property oracle, in which the relationship between $x$ and property $y$ is defined as, $y = \alpha_1 \exp(-(x-\mu_1)^2/2\sigma_1^2) + \alpha_2 \exp(-(x-\mu_2)^2/2\sigma_2^2)$. We choose the second Gaussian as the more desired mode by setting $\alpha_2 > \alpha_1$ (Figure 3). Here, separation is defined as the distance between the means of the two Gaussian modes ($\Delta\mu$). Larger values of $\Delta\mu$ are associated with higher separation. For each separation level, the train sets were generated by taking $N$ samples from the less desired mode, and taking $\rho N$ (imbalance ratio $\rho \leq 1$) samples from the more desired mode (see Appendix D.3).

For a fixed separation level, we compared the performance of PPGVAE and baseline methods for varying imbalance ratios. The relative improvements achieved by PPGVAE are consistently higher than all other methods and are robust to the imbalance ratio. Other methods achieve similar improvements when high-fitness samples constitute a larger portion of the train set (larger $\rho$). This happens at a smaller $\rho$ for lower separation levels (smaller $\Delta\mu$), indicating that the impact of imbalance is offset by a smaller separation between the optimum and the dominant region in training data (see Figure 3, Figure A4).

We then compared the relative improvement aggregated over all imbalance ratios, as the separation level increases. All methods perform well for low separation levels. PPGVAE stays robust to the degree of separation, whereas the performance of others drops by increasing separation (see Figure 3 top right). The difficulties encountered by other generative methods at higher separation levels is due to the difference between the reconstruction of low- and high-fitness samples. As $\Delta\mu$ increases, the more desired (high fitness) samples get mapped to a farther locality than the less desired ones. This makes the generation of more desired samples less likely under the VAE prior distribution $\mathcal{N}(0, I)$. This is exacerbated when more desired samples are rarely represented (small imbalance ratio) in the train set. For smaller $\Delta\mu$, more desired samples get mapped to a similar locality as less desired ones, regardless of their extent of representation in the train set. PPGVAE stays robust to the the imbalance ratio and extent of separation, as it always prioritizes generation and exploration of more desired samples over the less desired ones by mapping them closer to the latent space origin. Similar explanation can be provided for the rest of optimization tasks benchmarked in this study.

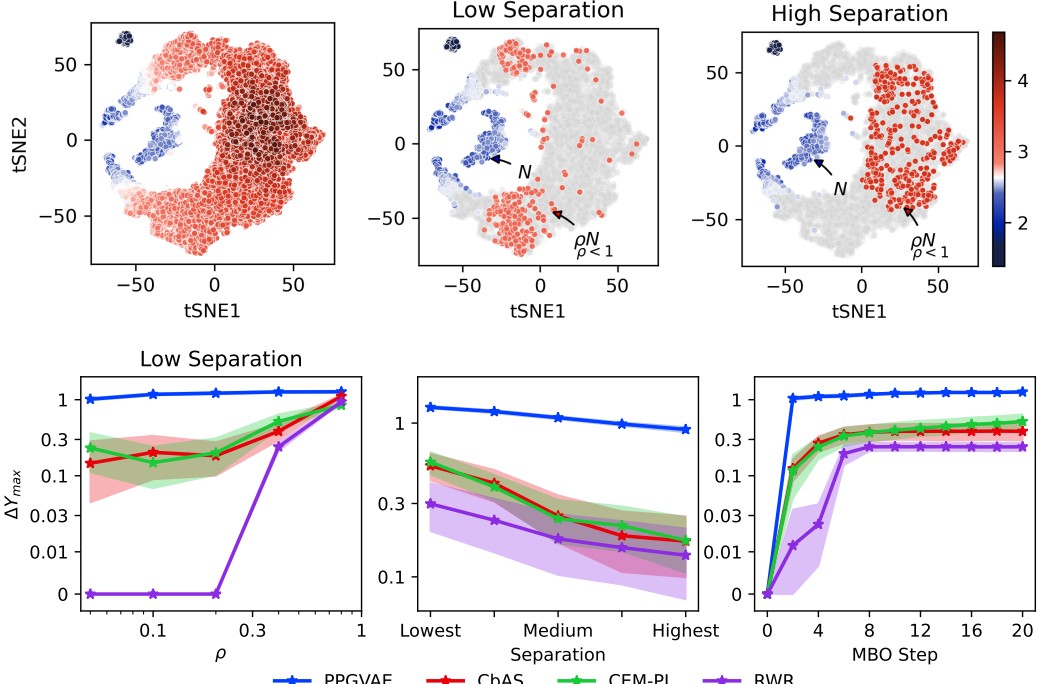

Figure 7: **Robustness to imbalance and separation in MBO for PINN.** The tSNE plot for the PINN-derived solutions colored with their property values (**top Left**). Blue and red color spectrum are used for lower and higher quality solutions, respectively. Top middle and right panels show train sets with low and high separation between the abundant less-desired and rare more-desired samples. PPGVAE achieves robust relative improvements, regardless of the imbalance ratio $\rho$ (**bottom Left**). Its performance stays robust to increasing separation (**bottom Middle**). It also converges in less number of MBO steps compared to other methods (**bottom Right**).

## 4.2 REAL PROTEIN DATASET

To study the impact of separation, real protein datasets with property (activity) measurements more broadly dispersed in the sequence space are needed. Such datasets are rare among the experimental studies of protein fitness landscape (Johnston et al., 2023), as past efforts have mostly been limited to optimization around a wild type sequence. We chose the popular AAV (Adeno-associated virus) dataset (Bryant et al., 2021) in ML-guided design (Sinai et al., 2021; Mikos et al., 2021). This dataset consists of virus viability (property) measurements for variants of AAV capsid protein, covering the span of single to 28-site mutations, thus presenting a wide distribution in sequence space.

The property values were normalized to $[0, 1]$ range. Threshold of 0.5 was used to define the set of low ($y < 0.5$) and high ($y > 0.5$) fitness (less and more desired respectively) mutants in the library. To define a proxy measure of separation, we considered the minimum number of mutated sites $s$ in the low-fitness training samples. A larger value of $s$ is possible only when the low-fitness samples are farther from the high-fitness region, i.e., at higher separation (see Figure 4). For a fixed separation $s$, the train sets were generated by taking $N$ samples from the low-fitness mutants containing at least $s$ number of mutations, and $\rho N$ ($\rho < 1$) samples from the high-fitness mutants (regardless of the number of mutations) (see Appendix D).

Given a fixed separation level, PPGVAE provides robust relative improvements, regardless of the imbalance ratio (Figure 4, Figure A5). CEM-PI is the most competitive with PPGVAE for low separation levels, however its performance decays for small imbalance ratios as separation increases (see Figure A5). The performance of other methods improve as the high-fitness samples represent a higher portion of the train set.

Next, we compared the performance of methods, aggregated over all imbalance ratios, as the separation level increases. PPGVAE, is the most robust method to varying separation. CEM-PI is

the most robust weighting-based generative method. Its performance is similar to PPGVAE for low separation levels and degrades as separation increases.

As it is desired to achieve improved samples with less sampling budget, we also studied the impact of varying sample generation budget per MBO step, given a fixed separation level. As expected, the performance increases by increasing the sampling budget for both CEM-PI and PPGVAE. Furthermore, on average PPGVAE performs better than CEM-PI for all sampling budgets (Figure 4). This is the direct result of prioritizing the generation and exploration of high-fitness samples relative to the low-fitness ones in the latent space of PPGVAE.

### 4.3 SEMI-SYNTHETIC PROTEIN DATASETS

Next, we used two popular protein datasets GB1 (Wu et al., 2016) and PhoQ (Podgornaia & Laub, 2015) with nearly complete fitness measurements on variants of four sites. However, these data sets exhibit a narrow distribution in sequence space (at most mutations), and are not ideal to study the effect of separation. We thus transformed the landscape of these proteins to generate a more dispersed dataset of mutants on which we could control for the separation of less and more desired variants. In this transformation, first a certain threshold on the property was used to split the dataset into two sets of low and high fitness mutants (see Appendix D). Second, a specific sequence of length $L$ was appended to high fitness mutants, while a random sequence of the same length was appended to the low fitness ones. Here, the length of the appended sequence determines the extent of separation. Higher separation is achieved by larger values of $L$ and makes the optimization more challenging. Note that the separation is achieved by changing the distribution of samples in the design space $\mathcal{X}$, while keeping the property values unchanged. For each separation level, train sets were generated by taking $N$ samples from the low fitness mutants and $\rho N$ samples from the high fitness mutants (see Figures 5 6, and Figure A3).

For a fixed separation level, the performance of all methods improve as high fitness samples constitute a higher portion of the train set, i.e., higher imbalance ratio (Figure 5 6). PPGVAE is more robust to the variation of imbalance ratio and it is significantly better than others when high fitness samples are very rare. Same observations hold for all separation levels studied (see Figures A6 A7). Similar as before, CEM-PI is the most competitive with PPGVAE for low separation; however as the separation increases its performance decays. Furthermore, the reduction of sampling budget does not affect the performance of PPGVAE as much as CEM-PI (Figures 5 6).

### 4.4 IMPROVING PINN-DERIVED SOLUTIONS TO THE POISSON EQUATION

Our method can also be used in continuous design spaces. We define the task of design as finding improved solutions given a training set overpopulated with low quality PINN-derived solutions. Details on train set generation and separation definition are covered in Appendix D.3. Similar conclusions hold for the robustness of PPGVAE to the separation level and imbalance ratio (Figure 7).

Common in all optimization tasks, PPGVAE achieves maximum relative improvement in less number of MBO steps than others (see Figures A9 A10 A11 A12). Characteristics of the latent space and sample generation have been studied for PPGVAE, and contrasted with prior generative approaches including (Gómez-Bombarelli et al., 2018) in Appendix A. Sensitivity of PPGVAE performance to the temperature is discussed in Appendix E.3 (see Figure A13).

## 5 CONCLUSION

We proposed a robust approach for design problems, in which more desired regions are rarely explored and separated from less desired regions with abundant representation to varying degrees. Our method is inherently designed to prioritize the generation and interpolation of rare more-desired samples which allows it to achieve improvements in less number of MBO steps and less sampling budget. As it stands, our approach does not use additional exploratory mechanism to achieve improvements, however it could become more stronger by incorporating them. It is also important to develop variants of our approach that are robust to oracle uncertainties, and study the extent to which imposing the structural constraint can be restrictive in some design problems.

## 6 ACKNOWLEDGEMENTS

This research was funded by Molecule Maker Lab Institute: an AI research institute program supported by National Science Foundation under award No. 2019897. This work utilized resources supported by 1) the National Science Foundation's Major Research Instrumentation program, grant No. 1725729 (Kindratenko et al., 2020), and 2) the Delta advanced computing and data resource which is supported by the National Science Foundation (award OAC 2005572) and the State of Illinois.

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

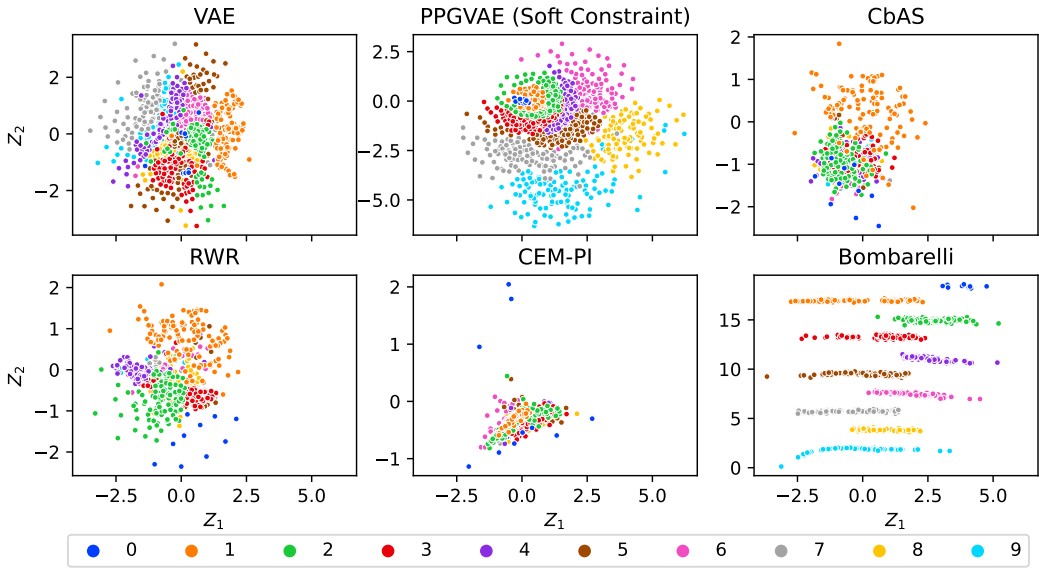

Figure A1: **Two-dimensional latent space of our PPGVAE and other methods trained on the toy MNIST dataset.**

## A Characteristics of the Latent Space and Sample Generation

To contrast PPGVAE with prior generative based methods, all methods were trained on the toy MNIST dataset of Figure 2. In weighting based methods, there is less distinction among the mapping regions of different classes, as the weighting becomes more extreme (Figure A1). Extreme weighting makes less samples contribute to the optimization, thus affecting the diversity in sample generation.

CEM-PI, CbAS and RWR have the most to least extreme weighting. In contrast to weighting based approaches, PPGVAE and Bombarelli have more distinct mapping of classes. Thus, sample generation using PPGVAE has more diversity. By increasing the standard deviation ($\sigma_s$) of the sampling prior $\mathcal{N}(0, \sigma_s I)$, PPGVAE is capable of generating all types of digits, whereas the least extreme weighting method (RWR) generates three types of digits only (Figure A2).

It is easily seen that the organization of points in the latent space of PPGVAE is significantly different from Bombarelli's. PPGVAE enforces the samples with higher properties to lie closer to the origin, whereas they are just separated by digit and ordered by property in Bombarelli's. By design, PPGVAE generates improved samples with higher probability, while exploration mechanisms need to be carefully designed for Bombarelli's. Finally, by sampling from VAE prior $\mathcal{N}(0, I)$, i.e., $\sigma_s = 1$, PPGVAE generates the top two highest property classes as well as their interpolations, more frequently than others. Optimization over the latent space of Bombarelli's barely produces samples from the top two highest property classes.

## B Related Work

Machine learning approaches have been developed to improve the design of DNA, molecules and proteins with certain properties. Early work in ML-assisted design (Gómez-Bombarelli et al., 2018), structures the samples in the latent space by joint training of a property predictor on the latent space. However, further systematic exploration mechanisms on the latent space are required to find improved samples. Variants of weighting based MBO techniques have also been proposed to facilitate the problem of protein design (Brookes & Listgarten, 2018; Gupta & Zou, 2019; Brookes et al., 2019). CbAS (Brookes et al., 2019) proposed a weighting scheme which prevents the search model from exploring regions of the design space for which property oracle predictions are unreliable. CbAS (Brookes et al., 2019) was built on an adaptive sampling approach (Brookes & Listgarten, 2018) that leverages uncertainty in the property oracle when computing the weights for MBO. Reward

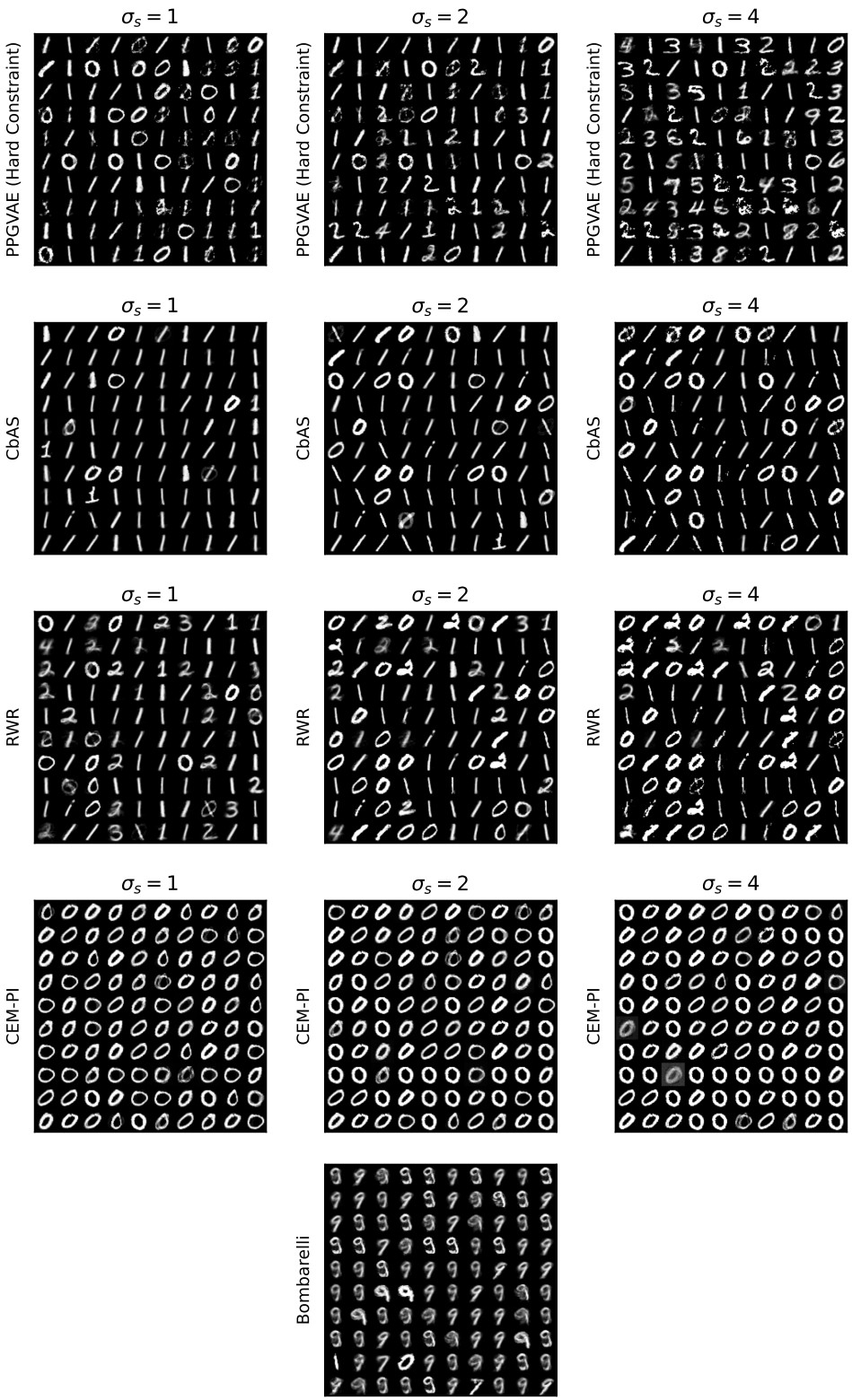

Figure A2: **Samples generated from different methods trained on the toy MNIST dataset (Figure A1).** For all methods except Bombarelli (Gómez-Bombarelli et al., 2018), normal sampling distribution $\mathcal{N}(0, \sigma_s I)$ with varying standard deviation $\sigma_s \in \{1, 2, 4\}$ was used. For Bombarelli, the optimization on the latent space was performed with a different starting point for each generated sample.

| Notation | Description |
|---|---|
| $\mathcal{X}$ | The design space |
| $x_i$ | Sample $i$ from the design space ($x_i \in \mathcal{X}$) |
| $y_i$ | Property value associated with $x_i$ |
| $w_i$ | Weight associated with $x_i$ |
| $Q_\theta$ | Encoder with parameters $\theta$ |
| $\mu_\theta^i$ | Encoded representation of $x_i$ |
| $\Pr(\mu_\theta^i)$ | Probability of the encoded representation w.r.t. the VAE pior |
| $\mathcal{L}_r$ | Relationship loss |
| $\lambda_r$ | Coefficient of the relationship constraint |
| $\tau$ | Temperature of relationship constraint |
| $p_\beta(y\|x)$ | Property oracle |
| $p_\theta(x)$ | Search model with parameters $\theta$ |
| $N_s$ | Number of samples generated per MBO step |
| $\rho$ | Relative proportion of more-desired to less-desired samples in train set |
| $N_{\text{eff}}$ | Effective sample size in weighted MLE |

Table A1: Mathematical notations used in the paper.

Weighted Regression (RWR) (Peters & Schaal, 2007) (benchmarked by CbAS) uses weights that do not take into account oracle uncertainty. Not requiring a differentiable oracle is common to these methods. On the other hand, a simple baseline proposed by (Trabucco et al., 2022) requires learning a differentiable property oracle that is used to find design samples by gradient ascent. In an effort to avoid failures due to imperfect oracles, (Kumar & Levine, 2020) proposed learning an inverse map from property space to design space and search for optimal property during optimization instead. Protein design has also been formulated as sequential decision-making problem (Angermueller et al., 2019), in which proximal policy optimization (PPO) (Schulman et al., 2017) has been used to search for design sequences. Recently, an evolutionary greedy approach has been shown to be competitive with prior algorithms (Brookes & Listgarten, 2018; Brookes et al., 2019; Angermueller et al., 2019). The latest work on protein design (Ren et al., 2022) (PEX), focuses on exploring the rugged landscape close to the wild-type sequence, to find high fitness sequences with low mutation counts.

In our benchmark for sequence design, we only included AdaLead as a baseline, selected over alternatives as it has been demonstrated to have superior performance to Dyna PPO (Angermueller et al., 2019). PEX (Ren et al., 2022) was not included, as it is specifically designed to perform a local search starting from the wild type, while our interest in the more challenging scenario of optima being distal to the wild type.

## C    EFFECTIVE SAMPLE SIZE IN WEIGHTED MLE

Common to all weighting-based generative methods is that weighted MLE could suffer from reduced effective sample size, which leads to estimation of parameters $\theta^{t+1}$ with higher variance and higher bias in extreme cases. This is inevitable in train sets with severe imbalance between the abundant undesired (e.g., zero property) and rare desired (e.g., nonzero property) samples. In contrast, PPGVAE does not use weights. Instead, it assigns higher probability to the more desired data points by restructuring the latent space. Thus, allowing for the utilization of all samples in training the generative model, regardless of the extent of imbalance between less and more desired samples.

| Dataset | Encoder Architecture |
|---------|----------------------|
| Protein | `indim` $\to 64 \to$ `LReLU` $\to 20$ |
| PINN | `indim` $\to 64 \to$ `LReLU` $\to 10$ |
| MNIST | `indim` $\to 512 \to$ `ReLU` $\to 256 \to$ `ReLU` $\to 2$ |
| GMM | `indim` $\to 64 \to$ `LReLU` $\to 64 \to$ `LReLU` $\to 2$ |

Table A2: **VAE architecture used for each benchmark dataset.** Encoder and decoder have symmetrical architecture. All generative based methods share the same architecture

In imbalanced datasets, sample weights ($w_i$) are typically uneven. As an example, assume a dataset of size 100 with five positive (desired property values), and 95 negative (undesired) samples. One weighting scheme can assign non-zero weights to the five desired samples and zero weights otherwise. Therefore, only five out of 100 samples contribute to the objective in Equation 2. This leads to higher bias and variance in the maximum likelihood estimator $\theta^{t+1}$ (MLE). Next, we present the same argument mathematically.

Representing the log-likelihood $\log(p_\theta(x_i))$ of the generative model with $l_i$, Equation 2 can be rewritten as maximizing

$$l = \sum_{i=1}^{K} w_i l_i. \tag{A8}$$

Assuming $\sum_{i=1}^{K} w_i = 1$ and i.i.d. samples, the effective sample size $N_{\text{eff}}$ (Kish & Frankel, 1974) can be defined such that

$$\text{Var}(l) = \frac{1}{N_{\text{eff}}} \text{Var}(l_i). \tag{A9}$$

According to Equation A8, $N_{\text{eff}} = (\sum_{i=1}^{K} w_i^2)^{-1}$. It can be proved that $1 \le N_{\text{eff}} \le K$ where the equality for the lower and upper bound holds at

$$N_{\text{eff}} = K, \quad \forall i, w_i = \frac{1}{K}, \quad \text{and}$$
$$N_{\text{eff}} = 1, \quad w_j = 1, \ w_{i \ne j} = 0. \tag{A10}$$

As mentioned earlier, uneven weights are expected in imbalanced datasets. As the weights become more uneven, $N_{\text{eff}}$ approaches its lower bound. Therefore, with imbalanced datasets, $N_{\text{eff}}$ tends to drop and $\text{Var}(l)$ increases (Owen, 2013). This in turn increases the estimation bias and variance of of the MLE $\theta^{t+1}$ (Firth, 1993; Ghaffari et al., 2022). Both estimation bias and variance are $\mathcal{O}(N_{\text{eff}}^{-1})$.

Our search model does not require weighting of the samples to prioritize the generation of some over the others. Instead, it directly uses the property values to restructure the latent space such that samples with better properties have a higher chance of being generated and interpolated. For this reason PPGVAE has $N_{\text{eff}} = K$, which makes it robust to the issues associated with parameter estimation in weighting-based MBO techniques.

## D  DETAILS OF THE EXPERIMENTS

### D.1  MBO SETTINGS AND IMPLEMENTATION DETAILS

We performed 10 rounds of MBO on the GMM benchmark and 20 rounds of MBO on the rest of the benchmark datasets. In all experiments temperature ($\tau$) was set to five for PPGVAE with no further tuning. We used the implementation and hyper-parameters provided by (Brookes et al., 2019), for CbAS, Bombarelli, RWR, and CEM-PI methods. The architecture of VAE was the same for all methods (Table A2). The formula to compute the weights for each weighting-based method are included in the Appendix of (Brookes et al., 2019).

| Dataset | Measured Property | Separation Criterion | Separation Quantities | Imbalance Ratio ($\rho$) |
|---|---|---|---|---|
| GMM | Bimodal Gaussian function | Distance between the two modes ($\Delta\mu$) | 4,6,8,10,12 | 0.05, 0.1, 0.2, 0.4, 0.8, 1 |
| PINN | -log(wMSE) | Property percentile for more desired samples | (30, 40), (40, 50), (50, 60), (60, 70), (70, 80) | 0.05, 0.1, 0.2, 0.4, 0.8 |
| GB1 | Folded protein enrichment | Length of the appended sequence | 3,4,5,8 | 0.0125, 0.025, 0.05, 0.1, 0.2, 0.4, 0.8 |
| PhoQ | Yellow fluorescent protein level | Length of the appended sequence | 3,4,6,8 | 0.0125, 0.025, 0.05, 0.1, 0.2, 0.4, 0.8 |
| AAV | Capsid viability | Minimum number of mutations for samples with less desired property ($y < 0.5$) | 6,8,10,12,15 | 0.0125, 0.025, 0.05, 0.1, 0.2, 0.4, 0.8 |

Table A3: **Settings used in train set generation for each benchmark dataset.**

The ground truth property oracle was used in all experiments, which is equivalent to limiting the search space to the sequences with measured properties in the protein datasets. The performance $\Delta Y_{\max}$ is reported with 95% bootstrap confidence interval.

In AdaLead, The oracle query size and the experiment batch size were both set to the number of samples generated per MBO step ($N_s$). This was to run AdaLead in a comparable setting to other weighting-based MBO approaches. Ground-truth oracle was used in all experiments, i.e., the search space was limited to the samples existing in the dataset.

For each setting of imbalance ratio and separation level, optimization was performed with 20 different random seeds. In AdaLead, the starting sequence was randomly selected from the initial train set for each seed. We noticed that AdaLead performance is highly dependant on where its search starts in the sequence space. This justifies the high variability in its performance. This is a common problem to all approaches based on evolutionary search, as it may not be known *a priori* where to start the search.

## D.2 PROTEIN DATASETS

**AAV:** The dataset contains variants of the protein located in the capsid of AAV virus along with their viability measurements. It consists of roughly 284K variants with single to 28 sites mutations. The properties were normalized into range $[0, 1]$. The dataset was obtained from (Dallago et al., 2021).

**GB1:** It contains the empirical fitness landscape for the binding of protein G domain B1 to an antibody. Enrichment of the folded protein bound to the antibody was defined as fitness. Fitness was measured for 149,361 sequences for variants of amino acids at four sites. The fitness was normalized to range $[0, 1]$ in this paper. The dataset is overpopulated with low or zero-fitness sequences. The sequences along with their properties were obtained from (Qiu & Wei, 2022).

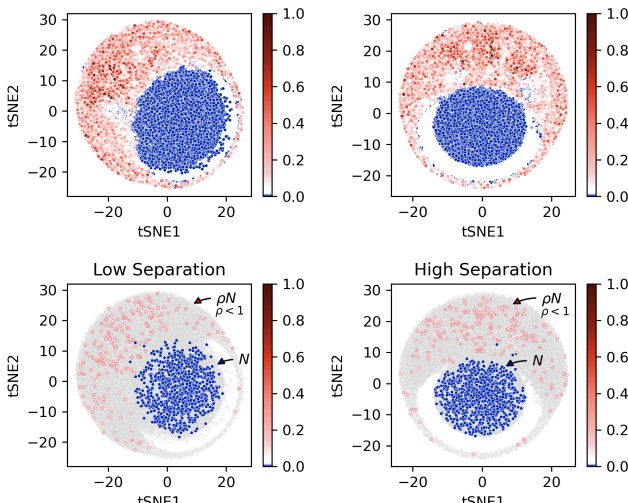

Figure A3: **GB1 transformed landscapes and train set examples.** Transformed landscapes with appended sequence of length three and six are shown in top left and top right panels, respectively. Examples of train sets taken from each landscape are shown in bottom left (low separation) and bottom right (high separation) panels, respectively.

**PhoQ:** It is a combinatorial library of four amino acid sites located at the interface between PhoQ kinase and sensor domains. Signaling of a two-component regulatory system was measured by yellow fluorescent protein (YFP) levels, i.e., fitness. The fitness landscape contains 140,517 measured sequences. The dataset is overpopulated with low or zero-fitness sequences. The fitness was normalized to range $[0, 1]$ in this paper. The sequences along with their properties were obtained from (Qiu & Wei, 2022).

### D.3 GENERATION OF TRAIN SETS WITH VARYING IMBALANCE AND SEPARATION

In each dataset, we specify the definition of less and more desired samples as well as the separation criterion in the design space $\mathcal{X}$. Table A3 includes the settings used to generate train sets in all benchmark datasets.

**GMM:** The property oracle is a bimodal GMM in which the second mode has a higher peak (higher mode) than the first mode (lower mode). Each mode is represented by a triplet $(\mu_i, \sigma_i, \alpha_i)$ where $\alpha_i$ determines the peak height and $i$ is the mode index. The triplet was set to $(0, 0.25, 1)$ and $(\mu_2, 1, 2.5)$ for the lower and higher modes, respectively. Mean of the second mode is variable as it determines the separation level.

Train sets consisted of less and more desired samples, which were taken from the two modes. $N$ samples were taken from the lower mode using $\mathcal{N}(0, 0.6)$ distribution. $\rho N$ samples representing the higher mode were taken uniformly from range $[\mu_2 + \sigma_2/2, \mu_2 + \sigma_2]$.

The separation in the design space was defined as the difference between the means of two modes $\Delta\mu$. Higher $\Delta\mu$ is associated with higher separation.

**AAV:** The entire AAV dataset consists of 284K samples (Bryant et al., 2021), which are split into two sets of "sampled" and "designed" variants (Dallago et al., 2021). We only used the sampled variants in this study. Threshold of $0.5$ on the viability scores was used to define the set of more desired ($> 0.5$) and less desired ($< 0.5$) variants.

Train sets consisting of less and more-desired samples were generated by, taking $N$ samples from the less-desired variants, whose mutation count was more than a specified threshold, and taking $\rho N$ samples from the more-desired variants, whose properties fall in the (5,10) percentile of the property values.

The separation is determined by the minimum number of mutations present in the less-desired samples. The separation of less and more-desired samples in the train set increases as the minimum number of mutations increases.

**GB1:** To study the impact of separation, the sequences of GB1 dataset were transformed to have a more dispersed coverage of the landscape in the design space. First, the property values were normalized into range $[0, 1]$. Threshold of $0.001$ on the property was used to define the set of less desired ($< 0.001$) and more-desired variants ($> 0.001$). Then, the sequences of less-desired variants were appended with random sequence of length $L$, whereas more-desired variants were appended with a specific sequence of the same length. Here, the length of the appended sequence specifies the degree of separation. Larger length is associated with higher separation.

Train sets were generated by sampling $N$ samples from the less-desired and $\rho N$ samples from the more-desired variants. The transformed landscape of GB1 associated with low and high separation, along with examples of its train sets are shown in Figure A3.

**PhoQ:** The same procedure as GB1 was used to generate the train sets.

**PINN:** We used a dataset of PINN-derived solutions to the Poisson equation for the potential of three point charges located diagonally on a 2D plane (Saleh et al., 2023). The solutions were pooled from a batch of PINNs trained with different seeds at different training epochs. Negative log weighted MSE (wMSE) between the PINN-derived solution and the analytical solution was used as the property value. Note that in practice, the exact average loss can be a proxy of the solution quality without the analytical solution. Higher quality solutions have higher properties.

In generating the train sets, $N$ low fitness samples were taken from $(0, 15)$ percentile of the property values, whereas $\rho N$ high fitness samples were taken from a specified range of property percentiles $(P_1, P_2) \mid P_1 \geq 30$. Higher values of $P_1$ are associated with higher separation levels. Note that in this case, separation of samples by property values is concordant with their separation in the design space $\mathcal{X}$. See Table A3 for the values of $(P_1, P_2)$ that have been tested.

In practice, as accurate solutions can be sporadic with stochastic optimizers, PPGVAE can interpolate higher-quality solutions given imbalanced sets of solutions.

**Toy MNIST:** This dataset was used to demonstrate the latent space of PPGVAE and other methods, as well as their sampling generation characteristics. The dataset has rare representation of zero digits relative to the other digits (imbalance ratio $\rho = 0.01$). Samples belonging to the digit class $C$ have property values distributed as $\mathcal{N}(10 - C, 0.01)$.

# E ADDITIONAL PLOTS

## E.1 STUDYING THE IMPACT OF IMBALANCE RATIO PER SEPARATION LEVEL

For a given separation, the performance improves as the imbalance ratio increases. This is expected as the more-desired samples constitute a higher portion of the train set. As separation level increases, the task of optimization becomes harder. Therefore, other methods become mostly ineffective for low imbalance ratios, and only show improvements for higher imbalance ratios. This behavior is consistently seen in all benchmark tasks.

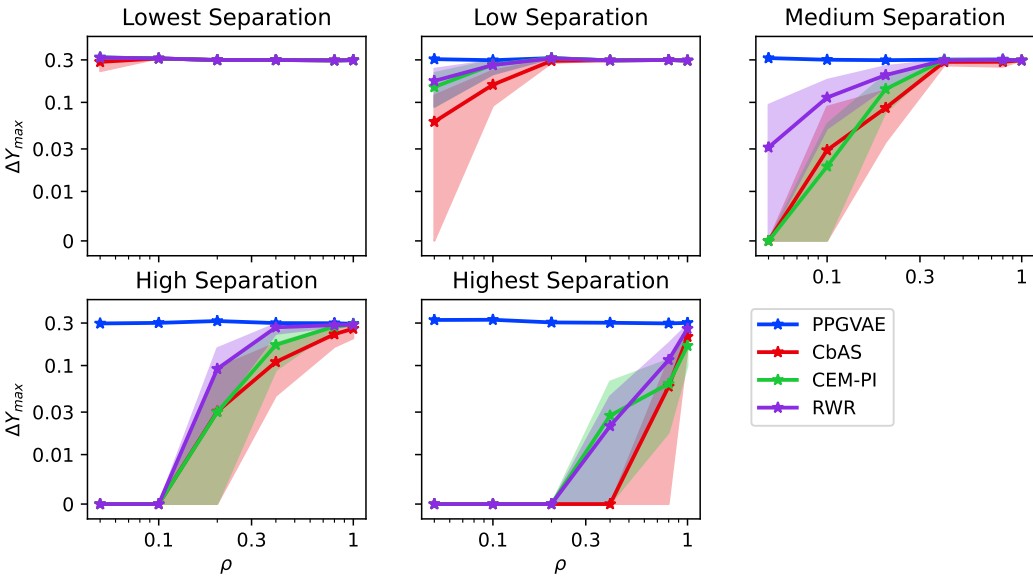

Figure A4: **Performance vs imbalance ratio for each separation level in the GMM benchmark.**

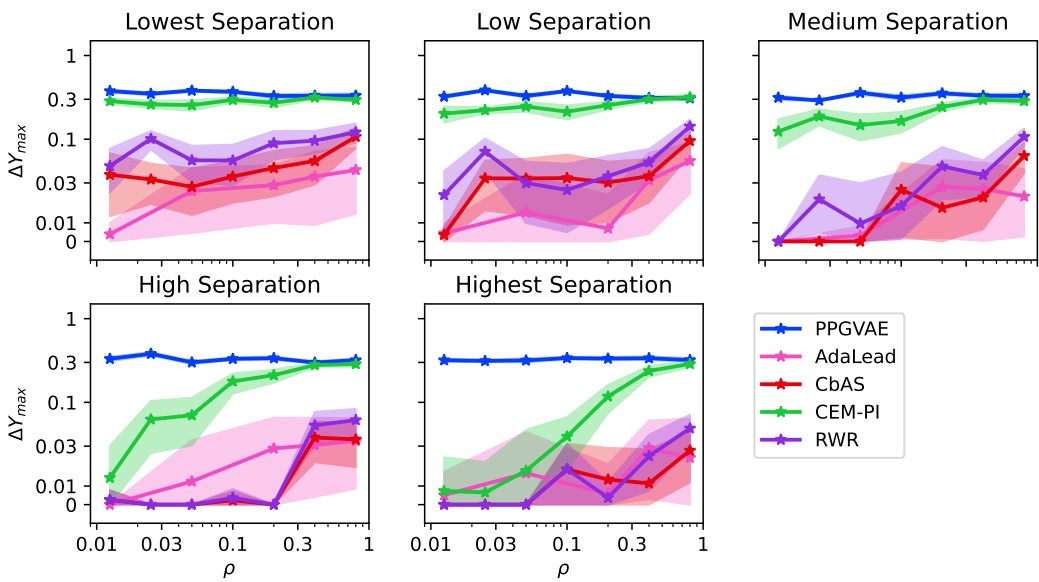

Figure A5: **Performance vs imbalance ratio for each separation level in the AAV benchmark.**

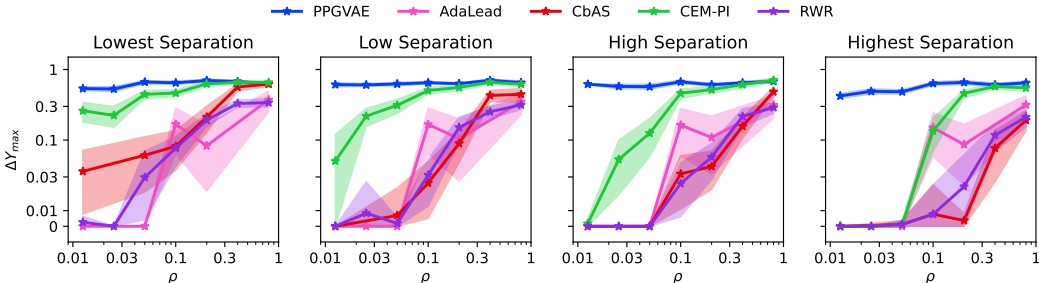

Figure A6: **Performance vs imbalance ratio for each separation level in the GB1 benchmark.**

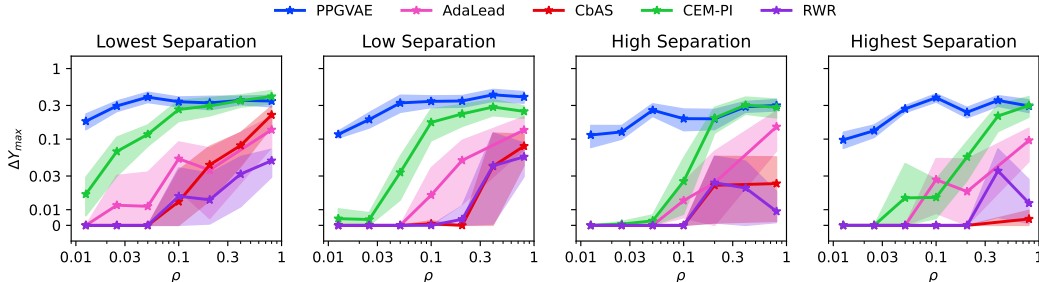

Figure A7: **Performance vs imbalance ratio for each separation level in the PhoQ benchmark.**

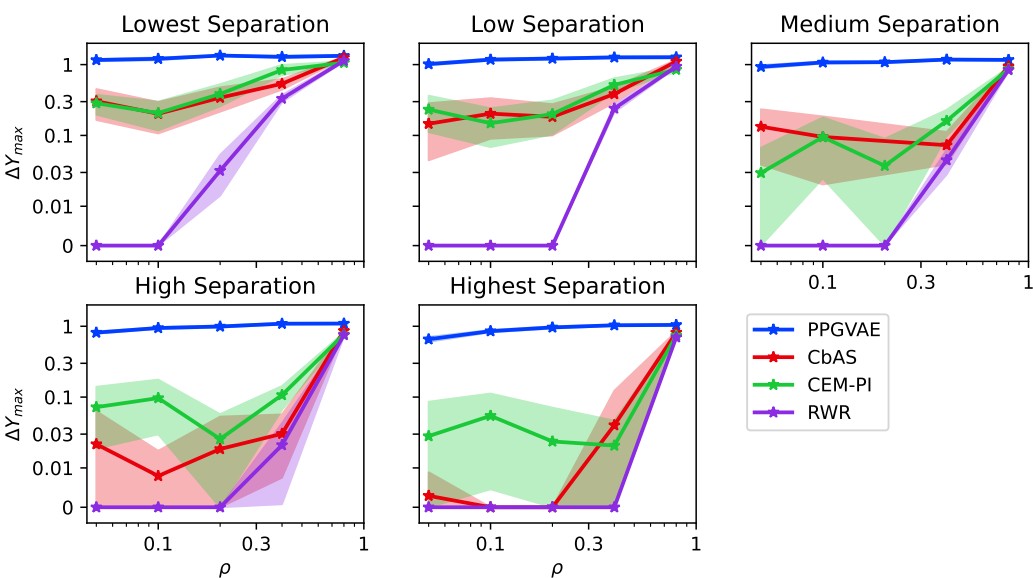

Figure A8: **Performance vs imbalance ratio for each separation level in the PINN benchmark.**

E.2    STUDYING THE CONVERGENCE RATE FOR DIFFERENT IMBALANCE RATIOS

For each benchmark task, we looked at the progress of each method as number of MBO steps increases. For higher imbalance ratios, all methods have faster convergence relative to the low imbalance ratios. Furthermore, our PPGVAE consistently has the highest convergence rate to the highest improvement. This is expected, as PPGVAE prioritizes the interpolation and generation of more desired samples, regardless of the extent of their representation in the train set.

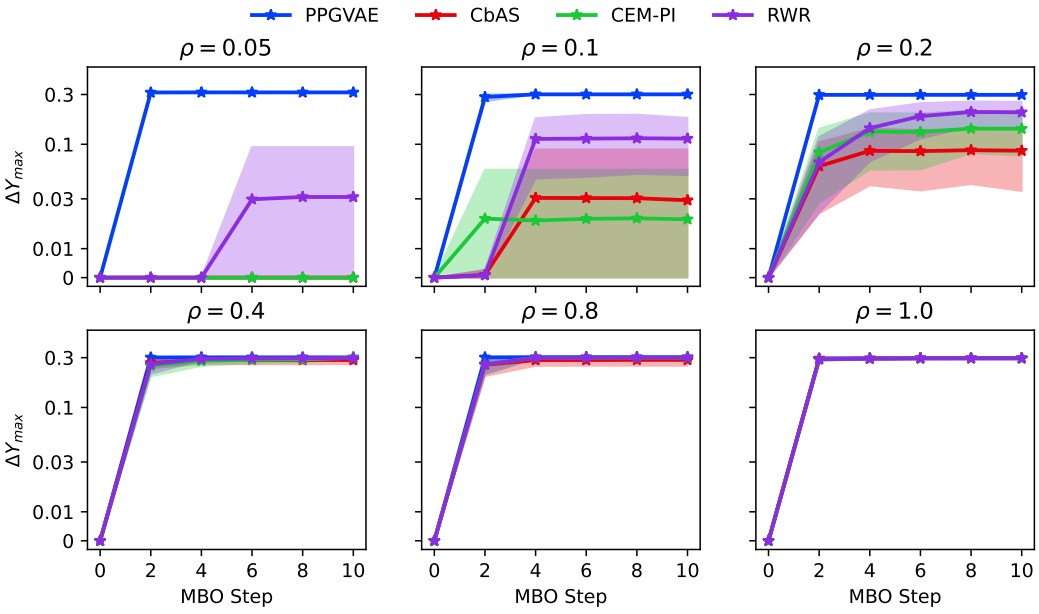

Figure A9: **Performance vs the number of MBO steps for different imbalance ratios in GMM benchmark. Separation level is set to medium.**

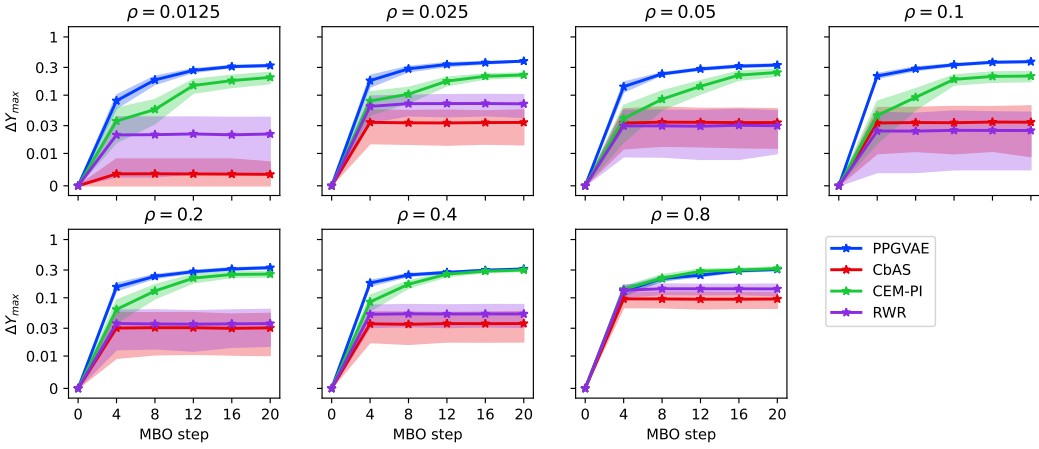

Figure A10: **Performance vs the number of MBO steps for different imbalance ratios in AAV benchmark. Separation level is set to low.**

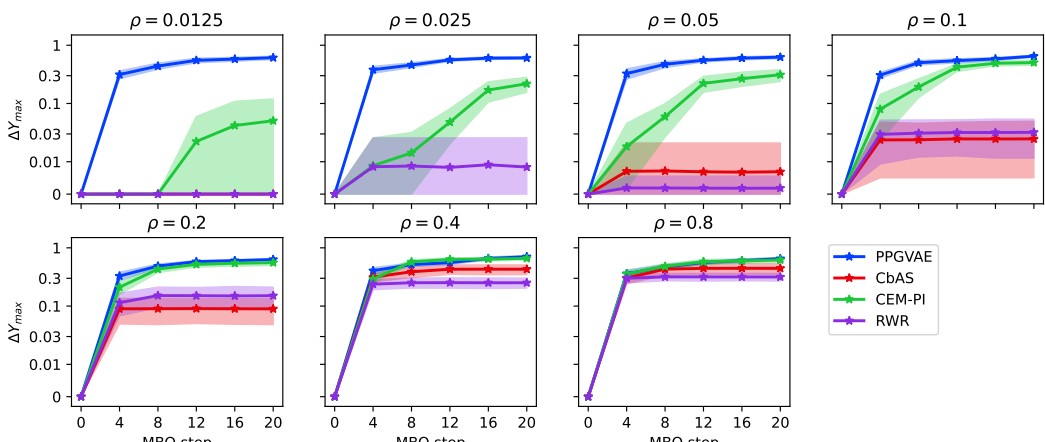

Figure A11: **Performance vs the number of MBO steps for different imbalance ratios in GB1 benchmark. Separation level is set to low.** CEM-PI is the most competitive with PPGVAE for higher imbalance ratios.

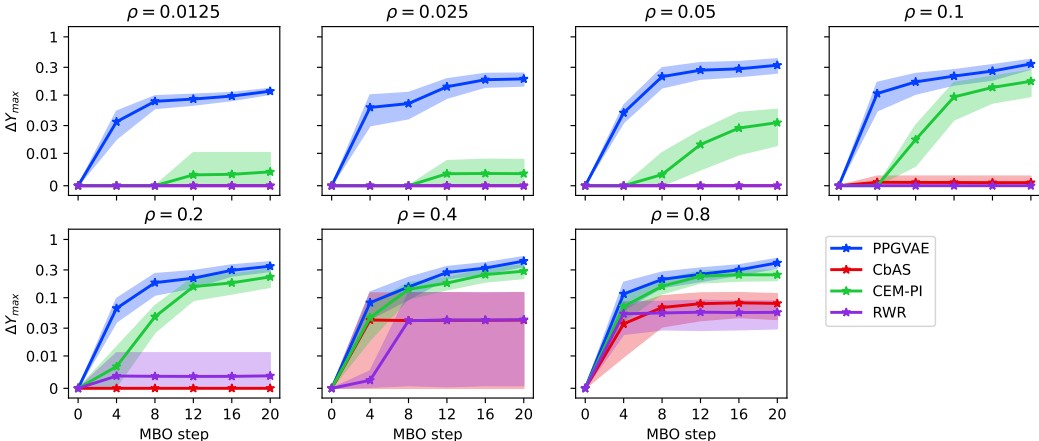

Figure A12: **Performance vs the number of MBO steps for different imbalance ratios in PhoQ benchmark. Separation level is set to low.** CEM-PI is the most competitive with PPGVAE for higher imbalance ratios.

### E.3    TEMPERATURE SENSITIVITY AND TRAINING ON HIGH FITNESS SAMPLES

To study the sensitivity of PPGVAE performance to the temperature, the GMM experiments with the lowest imbalance ratio and the highest separation (most challenging scenario) were performed with varying temperatures ($\tau$) Figure A13. The performance is almost the same for $\log_{10}(\tau) \in [-1, 1]$. Also, the sensitivity to temperature decreases as the number of MBO steps increases.

We also repeated the AAV experiments for CEM-PI in which only high fitness samples were used as the initial train set (CEM-PI/High) Figure A13. Aggregated performance over all imbalance ratios for PPGVAE is better than both CEM-PI and CEM-PI/High. This demonstrates the importance of including all samples in the training using PPGVAE. Furthermore, CEM-PI/High has better performance than CEM-PI for higher separation, showing that filtering the samples might be beneficial for weighting based approaches as the optimization task gets harder by separation criterion.

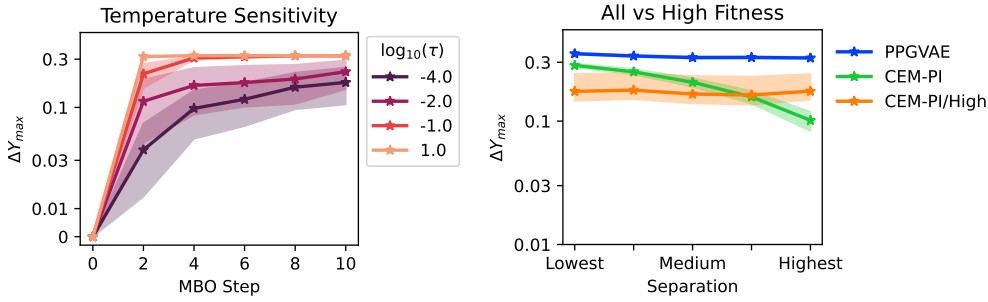

Figure A13: **Impact of varying temperature in GMM benchmark (Left) and performance comparison between training on all vs high fitness samples for CEM-PI (Right).**

