# OpenReview forum: "Robust Model-Based Optimization for Challenging Fitness Landscapes"
_ICLR.cc/2024/Conference — ICLR 2024 poster_

### Official Review · Reviewer_7H3r · 2023-10-27

**Soundness:** 2 fair
**Presentation:** 2 fair
**Contribution:** 2 fair
**Rating:** 6
**Confidence:** 1

**Summary:**

Authors propose to use a Variational Autoencoder (VAE) to model a protein engineering task. Obtained model would allow scientists from the area to better know where to look for promising compounds in such difficult search space. They argue that their approach is more effective and efficient than other normally attempted approaches, such as RL guided searches or evolutionary optimization. Authors provide some experimental assessments that include experiments with artificial datasets to properly evaluate their approach.

**Strengths:**

Unfortunately, Bioinformatics is far outside my scope areas. I can't really assess quality of the paper. For the untrained eye, everything seems to fit. The proposed approach seems correct for the task at hand (as far as I could understand it), but I'm completely unfamiliar with the related work, can't really say if this has been attempted before for this particular domain, or to what extent. All I can say is that presentation of the paper is good, and the artificial dataset experiments somewhat seem to validate what authors attempted to do.

**Weaknesses:**

As I already mentioned, I can't really assess the draft.
The only one thing that I'd like to bring forward, it's that for the given bioinformatics task described -as far as I could understand it- VAEs seem like a natural thing to try. I'm surprised nobody has done it before; but since I'm unfamiliar with the Related Work, surely there are things that I'm missing.

**Questions:**

I'm leaning a bit towards rejection of the paper, just for the reason I mentioned in the 'weaknesses' section, about novelty and contribution.
But if other reviewers go with acceptance, I won't argue against.
AC has been alerted to seek another opinion if needed.

---

> ### Author Response · Authors · 2023-11-22
> **Response to Reviewer 7H3r**
>
> We thank the reviewer for their feedback and time.
>
> **Question 1:** The only one thing that I'd like to bring forward, it's that for the given bioinformatics task described -as far as I could understand it- VAEs seem like a natural thing to try. I'm surprised nobody has done it before; but since I'm unfamiliar with the Related Work, surely there are things that I'm missing
>
> **Response:**
> The novelty of our work lies in looking into the less explored problem of separation in the design space, in which the desired optimum is in a region that is not only poorly represented in training data, but also relatively far from the highly represented low-fitness regions.
>
> Prior work, such as CbAS included in the benchmark, have used VAE for the task of sequence generation. Our work is not novel in proposing to use VAEs for the first time. Rather, it addresses the overlooked problem of separation using a modified VAE.

---

> > ### Comment · Reviewer_7H3r · 2023-11-22
> >
> > Thank you for your clarification. I'll revisit your modified draft in the future under a new light considering your answer to my concern. In the meanwhile I'll update my score to 6.
> > Regards

---

> > > ### Author Response · Authors · 2023-11-23
> > >
> > > Dear Reviewer 7H3r,
> > >
> > > We appreciate your valuable feedback and follow-up! We are glad to hear that our response has addressed your concerns, and we are grateful for your willingness to increase the score to 6. Thanks a lot once again!

---

### Official Review · Reviewer_R8Dk · 2023-10-31

**Soundness:** 3 good
**Presentation:** 3 good
**Contribution:** 3 good
**Rating:** 8
**Confidence:** 3

**Summary:**

This paper identifies the scenarios where the desired optimum is in a region that is not only poorly represented in training data but also relatively far from the highly represented low-fitness regions for model-based optimization for sequence-function landscapes, named separation, and proposes a new method using a variational auto-encoder to explicitly structure the latent space by property values of the sequences such that more desired samples are prioritized over the less desired ones and have higher probability of generation. Empirical studies on three real and semi-synthetic protein datasets show the robustness of the proposed method.

**Strengths:**

1.	This paper identifies the problem of separation for model-based optimization for sequence-function landscapes.
2.	A robust method based on a variational auto-encoder is proposed to solve the separation and sparsity problems.
3.	Empirical studies are conducted on three real and semi-synthetic protein datasets to study the effectiveness of the proposed method.

**Weaknesses:**

1.	A clear description of the robustness concept needs to be provided for friendly reading.
2.	A description of the organization of this paper as well as a Conclusion session should be added.

**Questions:**

What is the maximum dimensionality examined in the experiments? How will the proposed method perform when increasing the dimensionality of an optimization problem?

---

> ### Author Response · Authors · 2023-11-22
> **Response to Reviewer R8Dk**
>
> We thank the reviewer for understanding the strengths and contribution of our work to the field of protein design.
>
> **Question 1**
> A clear description of the robustness concept needs to be provided for friendly reading.
>
> **Response:**
> Thanks for your suggestion. We have now added the following sentence to the Introduction, where the central goal of the work is introduced: “ A robust algorithm should have consistent performance under varying separation and imbalance.”
>
>
> **Question 2**
> A description of the organization of this paper as well as a Conclusion session should be added.
>
> **Response:**
> Description of the organization of the paper has been added. The section previously titled “Discussion” is meant to be our conclusion section and has been re-titled following the reviewer’s suggestion.
>
>
> **Question 3**
> What is the maximum dimensionality examined in the experiments? How will the proposed method perform when increasing the dimensionality of an optimization problem?
>
> **Response:**
> The input dimension changes from one to 2500 in our experiments. PPGVAE did well in all benchmarks. PINN benchmark has the maximum input dimension of 2500. For protein datasets, AAV has the largest input dimension with sequences of length 28 and encoding of each amino acid with a 20-dimensional vector. The dimension of the latent space ranges from 2 to 20 in our experiments. This is included in the Appendix Table A2.

---

> > ### Comment · Reviewer_R8Dk · 2023-12-05
> >
> > Thank you for your response and the modifications you made. They have helped clarify my questions.

---

### Official Review · Reviewer_soEo · 2023-10-31

**Soundness:** 3 good
**Presentation:** 3 good
**Contribution:** 3 good
**Rating:** 5
**Confidence:** 2

**Summary:**

Note: I am not familiar with the field of protein design, so this review is largely from the perspective of generative modelling.

**Problem Setting**

We are considering the problem of protein design, in which data consists of sparse examples of high-fitness designs. Due to the nature of the optimization problem, it is desirable to learn a model that can judge the fitness of specific sequences.

**Novel Idea**

This work proposes the learning of a weighted VAE such that high-fitness samples are weighted in the probability distribution. The intuition is that exploration in VAE space is more likely to result in succesful generations with this bias in place. However, all samples can still be used in training the model.

The PPGVAE is structured so that designs with higher fitness are generated closer to the origin. This means they will be more likely to be sampled when using a normal distribution prior. This is implemented in the form of a constraint between the ratio of log probabilities vs. fitness, and this constraint is relaxed to a weighted penalty.

**Findings**

Experiments showcase that the PPGVAE model can correctly separate between two modes. The modes are defined as the probabilities represented by a two-mode Gaussian mixture model. On protein datasets, the method showcases that the PPGVAE can separate between low and high fitness examples, and that separated models lead to faster convergence when using MBO.

**Strengths:**

This paper presents a useful method for learning an exploration prior for model based optimization. In the area of protein design, there are sparse examples of high fitness designs, and many low fitness designs. A good prior should bias towards the space of high-fitness designs, while still making use of all examples. This work presents a simple and clean objective that learns a variational auto-encoder. The methodology is clear and theoretical justification is provided. It may provide significance in the specific problem domain, although the method itself is domain-agnostic.

**Weaknesses:**

The figures of the paper were confusing in terms of what message they were trying to convey. A more informative caption describing why the results are important would improve the clarity here.

The experiments do showcase the the proposed architecture helps in terms of separating high-fitness examples. It would be insightful to include some experiments on what happens if the VAE is simply trained on only the high-fitness examples and the low-fitness examples are dropped altogether.

It is unclear the difference between the hard constraint and the soft constraint. Are these just different hyperparameters on the penalty? Or is it a true constraint using i.e. a Lagrange multiplier?

**Questions:**

See above for a list of questions.

Is the intent of this work to be applicable to other domains, or only protein design? If the answer yes, as implied by the title, it would be greatly strengthening to apply this method on more classical generative modelling tasks larger than MNIST.

---

> ### Author Response · Authors · 2023-11-22
> **Response to Reviewer soEo**
>
> We thank the reviewer for recognizing the novelty of PPGVAE and proposing new analysis to improve the work.
>
> **Question 1**
> The figures of the paper were confusing in terms of what message they were trying to convey. A more informative caption describing why the results are important would improve the clarity here
>
> **Response:**
> Thanks for the suggestion. We have now updated the captions to be more informative. We have also clarified "robustness" in the main text.
>
> **Question 2**
> It would be insightful to include some experiments on what happens if the VAE is simply trained on only the high-fitness examples and the low-fitness examples are dropped altogether.
>
> **Response:**
> We thank the reviewer for the suggestion. We have repeated AAV experiments using CEM-PI and the high-fitness samples (“CEM-PI/High”) as the initial training set. The aggregated performance over all imbalance ratios vs the separation level is plotted in Appendix Figure A13 for PPGVAE and CEM-PI (both trained on all samples), as well as CEM-PI/High.
>
> Aggregated performance over all imbalance ratios for PPGVAE is better than both CEM-PI and CEM-PI/High. This demonstrates the importance of including all samples in the training (as in PPGVAE). Furthermore, CEM-PI/High has better performance than CEM-PI for higher separation, showing that filtering the samples might be beneficial for weighting based approaches as the optimization task gets harder by separation criterion.
>
> In practice it is not known where to set the threshold for defining low and high-fitness samples. The weighting-based methods may take advantage of filtering the samples as they suffer more from the overabundance of low-fitness samples; however this comes at the expense of losing the chance of generating samples that are interpolations of low and high fitness samples, which in turn may hurt the overall performance of MBO.
>
> **Question 3**
> It is unclear the difference between the hard constraint and the soft constraint. Are these just different hyperparameters on the penalty? Or is it a true constraint using i.e. a Lagrange multiplier?
>
> **Response:**
> In the hard constraint, $\lambda_r$ (coefficient of the relationship loss, Equation 5) is set to a relatively large constant throughout the training. While in the soft constraint, $\lambda_r$ gradually decreases as the training goes on. We have added this clarification to the text.
>
> **Question 4**
> Is the intent of this work to be applicable to other domains, or only protein design? If the answer yes, as implied by the title, it would be greatly strengthening to apply this method on more classical generative modelling tasks larger than MNIST.
>
> **Response:**
> The reviewer is correct in that our method is domain agnostic. However, it was originally motivated to improve MBO for the problem of protein design in challenging real world examples. We mainly used the MNIST example to showcase the characteristics of PPGVAE compared to other approaches. Our intention is to improve upon the current model and apply it for image generation for larger image datasets in future.

---

> > ### Comment · Reviewer_soEo · 2023-11-22
> >
> > Thank you for the response and the clarifications, which look to strengthen the paper. Given that the details above were fixed in the revision, I would update my score to a 6.

---

> > > ### Author Response · Authors · 2023-11-22
> > >
> > > Dear Reviewer soEo,
> > >
> > > We appreciate your valuable feedback and follow-up! We are glad to hear that our response has addressed your concerns, and we are grateful for your willingness to increase the score to 6. We notice that the score has not been updated accordingly. Would you mind helping update the score at your earliest convenience? Thanks a lot once again!

---

### Official Review · Reviewer_e46j · 2023-11-01

**Soundness:** 3 good
**Presentation:** 3 good
**Contribution:** 3 good
**Rating:** 6
**Confidence:** 3

**Summary:**

The paper addresses a crucial challenge in protein design, which involves optimization on a fitness landscape. The main concern in current model-based optimization methods is the sparsity of high-fitness samples in training datasets and the separation problem - wherein the desired optimum is situated in a region that is poorly represented and far from low-fitness areas. This paper pinpoints that existing tools do not efficiently handle this separation problem in the design space.

The authors introduce a new method using Property-Prioritized Generative Variational Auto-Encoder (PPGVAE). This VAE's latent space is structured by the fitness values of the samples, ensuring higher prioritization and generation probability for more desired sequences. This new method aims for better results with fewer optimization steps, which is particularly valuable for sequence design problems.
A comparative advantage of this approach over prior methods is demonstrated via extensive benchmarks on real and semi-synthetic protein datasets.

The PPGVAE proves to be superior in robustly finding improved samples, regardless of the imbalance between low- and high-fitness samples and the degree of their separation in the design space. The authors further extend the versatility of their method by testing it on continuous design spaces, showcasing its efficacy on physics-informed neural networks (PINN).

**Strengths:**

1. The paper recognizes the less-explored challenge of "separation" in protein design space, which is a significant departure from recent studies that have mostly focused on the sparsity of high-fitness samples. And The PPGVAE proposed in the paper is an effective approach to tackling the separation issue in model-based optimization.

2. The paper does not just present a theoretical model but comprehensively demonstrates its effectiveness through extensive benchmarking on real and semi-synthetic protein datasets. Beyond just protein datasets, the paper further validates the model on continuous design spaces, exemplified with physics-informed neural networks (PINN).

**Weaknesses:**

1. Limitation on the conducting experiments exclusively on the real protein dataset of AAV. While this might result in high accuracy and performance metrics within this context, the method may not readily translate to other proteins or protein datasets, especially if they possess distinct characteristics or functionalities.

**Questions:**

1. How easily can PPGVAE be extended to prioritize or balance multiple properties simultaneously? Would this require a significant alteration to the existing framework?
2. Could you provide insights into the sensitivity of the model's performance to changes in the temperature in the relationship loss?

---

> ### Author Response · Authors · 2023-11-22
> **Response to Reviewer e46j**
>
> We thank the reviewer for appreciating the novelty of our work and its extensive benchmark.
>
> **Question 1 (Weakness):**
> Experiments exclusively on the real protein dataset of AAV, hence the method may not readily translate to other proteins or protein datasets.
>
> **Response:**
> The choice of AAV dataset was mainly due to its wide mutational coverage (samples dispersed in the sequence space) which allows us to test the effectiveness of our approach. Most other experimental protein datasets do not have such wide coverage; this is because typically protein engineering seeks an optimum close to the starting point (wild type). However, there are protein engineering challenges in which the optimum is expected to lie far from the starting sequence, as explained in the paper, e.g., design of an enzyme for an unnatural target substrate.
>
> We agree with the reviewer that the advantages of our method may not readily translate to the goals of other protein engineering tasks, e.g., when the designer can safely assume the existence of an optimum close to the starting sequence. We strongly believe that our method will open up the opportunity to address the more general but harder engineering problem where this assumption cannot be made. With our method in hand researchers will be more open to embarking on the harder problem, thus leading to more data sets of wide mutational coverage.
>
> **Question 2**
> How easily can PPGVAE be extended to prioritize or balance multiple properties simultaneously? Would this require a significant alteration to the existing framework?
>
> **Response:**
> Thanks for this important question/suggestion. There are two ways to approach this. The simpler approach will be to combine multiple properties into a single property with a proper transformation. A second possible approach may be through modified training of PPGVAE: assign a subset of dimensions in the latent space to each property and enforce the relationship loss for each subset of latent dimensions (property). This idea will require future implementation and testing.
>
> **Question 3**
> Could you provide insights into the sensitivity of the model's performance to changes in the temperature in the relationship loss?
>
> **Response:**
> To study the sensitivity of performance to temperature, we reran the GMM experiments in the lowest imbalance ratio and highest separation setting (the most challenging scenario), with varying temperature values. The results are included in Appendix E3. Based on this analysis, the performance is almost the same for $\log_{10}(\tau)\in [-1,1]$. Also, the sensitivity to temperature decreases as the number of MBO steps increases (see Figure A13).
>
> In general, at very high temperature, even small differences in property values will be reflected in the ordering of samples in the latent space. On the other hand, at very low temperature all samples will have the same probability of generation and lie on the same sphere. As an added constraint to the reconstruction loss, both extremes damage a good reconstruction of samples and affect the quality and diversity of generated samples. Thus, it is important to set it to a reasonable value.
>
> We should point out that we performed all the experiments in the paper with a fixed temperature, as mentioned in the paper. Further tuning of the temperature may improve the results, however we were able to show the usefulness of our approach without tailored tuning for each design scenario.

---

### Author Response · Authors · 2023-11-22
**General Response**

We thank all reviewers for their time and effort and for their constructive feedback. We are excited that the reviewers find our work to be “a significant departure from recent studies” (e46j), and that we do “not just present a theoretical model but comprehensively demonstrates its effectiveness” (e46j). Further, we are happy that reviewers recognize that we present “a useful method” (soEo) with “ a simple and clean objective” (soEo) that is “robust” (R8Dk) and the “presentation of the paper is good” (7H3r).

We have modified the main text of the paper as requested by the reviewers, and added the suggested analyses (temperature sensitivity and training on high-fitness data) to the end of the supplementary section. All modifications are in blue for convenient lookup.
We answer individual reviewer questions below.

---

### Meta-Review · Area_Chair_N2Ws · 2023-12-11

**Metareview:**

The paper focuses on model based optimization of challenging fitness landscapes. It identifies two challenges in this area, namely what they call "sparsity" in high-fitness samples in training datasets and the "separation" problem, where the desired optimum is in a poorly represented region and distant from highly represented areas. To address these challenges, the authors propose a new model called Property-Prioritized Generative Variational Auto-Encoder (PPGVAE), which structures the latent space by the fitness values of the samples, placing high fitness samples closer to the origin and giving higher generation probability to high fitness samples. The effectiveness of the proposed method is demonstrated through benchmarking on real and semi-synthetic protein datasets and on a continuous design spaces problem for finding solutions to the Poisson equation. The reviewers appraised the paper for identifying and addressing an overlooked challenge, that the metrology is clear and theoretical justification, the extensive empirical evaluation, and that the proposed methodology in principle is generally applicable (i.e., domain-agnostic). However, the reviewers problematized that the method was not evaluated on more diverse datasets with different characteristics, making the empirical conclusions dataset specific.

**Justification For Why Not Higher Score:**

The justification for not awarding a higher score is grounded in the reviewers' concerns about the paper's generalizability to other datasets.

**Justification For Why Not Lower Score:**

The justification for not having a lower score is based on the fact that all the reviewers recognize the paper’s contributions and significance, and they all recommend accepting the paper.

---

### Decision · Program_Chairs · 2024-01-16

Accept (poster)